# WHEN CAN ISOTROPY HELP ADAPT LLMS' NEXT WORD PREDICTION TO NUMERICAL DOMAINS?

## ABSTRACT

Recent studies have shown that vector representations of embeddings learned by pre-trained large language models (LLMs) are effective in various downstream tasks in numerical domains. Despite their significant benefits, the tendency of LLMs to hallucinate in such domains can have severe consequences in applications like finance, energy, retail, climate science, wireless networks, synthetic tabular generation, among others. To guarantee prediction reliability and accuracy in numerical domains, it is necessary to have performance guarantees through explainability. However, there is little theoretical understanding of when pre-trained language models help solve numeric downstream tasks. This paper seeks to bridge this gap by understanding when the next-word prediction capability of LLMs can be adapted to numerical domains through the lens of isotropy. Specifically, we first provide a general numeric data generation process that captures the core characteristics of numeric data across various numerical domains. Then, we consider a log-linear model for LLMs in which numeric data can be predicted from its context through a network with softmax as its last layer. We demonstrate that, in order to achieve state-of-the-art performance in numerical domains, the hidden representations of the LLM embeddings must possess a structure that accounts for the shift-invariance of the softmax function. We show how the isotropic property of LLM embeddings preserves the underlying structure of representations, thereby resolving the shift-invariance problem problem of softmax function. In other words, isotropy allows numeric downstream tasks to effectively leverage pre-trained representations, thus providing performance guarantees in the numerical domain. Experiments show that different characteristics of numeric data could have different impacts on isotropy.

## 1 INTRODUCTION

Large language models (LLMs) have demonstrated broad success in adapting to various downstream tasks in numerical domains, such as finance Garza & Mergenthaler-Canseco (2023); Yu et al. (2023), energy Gao et al. (2024), retail, climate science Jin et al. (2024), wireless communications Xu et al. (2024), synthetic tabular generation Dinh et al. (2022); Borisov et al. (2023); Xu et al. (2024), among others. For many of these numeric downstream tasks, training a linear classifier on top of the hidden-layer representations generated by the pre-trained models have already shown near state-of-the-art performance Jin et al. (2024); Ansari et al. (2024). Despite their significant benefits in the numerical domains, the LLMs' tendency to hallucinate can have serious consequences in numeric applications. To ensure prediction reliability and accuracy in numerical domains, a promising approach would be to instill performance guarantees through explainability. Although recent empirical studies Jin et al. (2024); Nie et al. (2023); Liu et al. (2024) demonstrate the benefits of vector representations of embedding learned by LLMs in various numeric downstream tasks, there is little theoretical understanding of their empirical success. Thus, a fundamental question arises: *"when the next-word prediction capability of LLMs can be effectively adapted to numerical domains?"*

The main contribution of this paper is to answer this question through the lense of *isotropy*. Isotropy refers to the geometric property where vector representations in the embedding space are uniformly distributed in all directions, a characteristic critical for maintaining the expressiveness of the embedding space Arora et al. (2016); Mu & Viswanath (2018). In particular, we first introduce a general data generation process for numerical domains and consider a log-linear model of LLMs for nu-

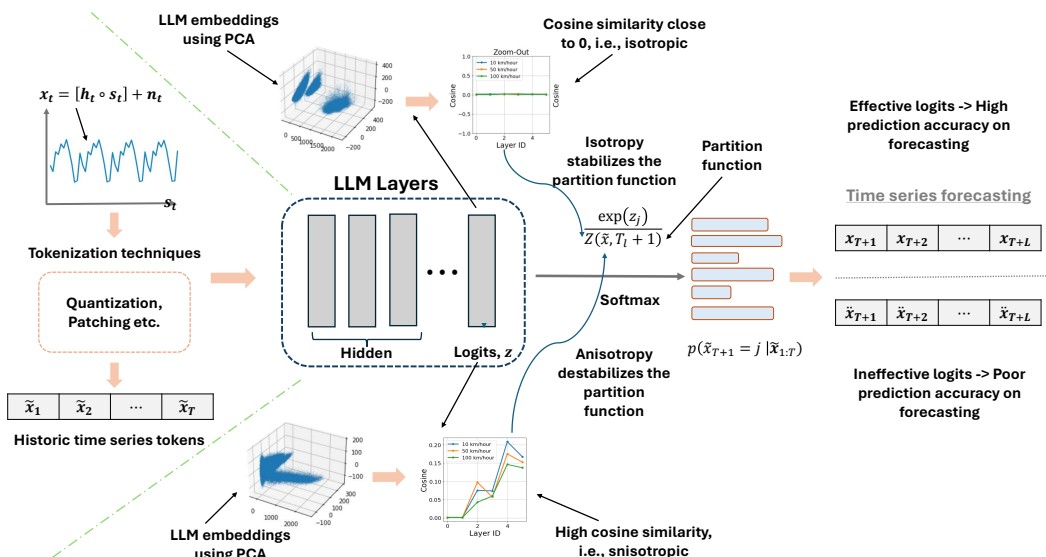

Figure 1: High-level illustration on when pre-trained language models help solve numeric downstream tasks: the hidden representations of LLM embeddings must exhibit structure to address the shift-invariance problem of the softmax function. The isotropic property of LLM embeddings preserves the underlying structure of the representations by approximating the partition function with a constant for different samples, thereby resolving the shift-invariance problem of softmax function and providing a performance guarantee.

meric downstream tasks. To achieve state-of-the-art performance in numerical domains, we show that the hidden representations of LLMs must exhibit *a structured form* that accounts for the shift-invariance of the softmax function (i.e., the softmax output remains unchanged when all logits are shifted by a constant). Without such structure, the model can shift the logits while keeping the training loss unchanged thereby rendering the logits ineffective for downstream tasks. In the worst case, the model can rapidly shift the logits for unseen numeric data, thus resulting in poor performance. We further show that, when isotropy preserves the structure in LLM representations, it resolves the shift-invariance problem, and thus ensuring that the logits are useful for numerical downstream tasks (see Figure 1 for a high-level illustration on when pre-trained language models help solve numeric downstream tasks). In summary, our key contributions include:

- We first provide a general numeric data generation process that captures the core characteristics of numeric data across various domains. Then, we consider a log-linear model for LLMs where numeric data are predicted from contexts through a network with softmax as the final layer, along with the cross-entropy-based loss function.

- We then showcase the role of isotropy in adapting LLMs to numerical data. In particular, we derive a theorem to investigate why hidden representations must exhibit structure to address the shift-invariance problem of the softmax function. The theorem indicates that without structural constraints, the log-linear model can shift the logits for any numeric data sample without affecting the pre-training loss.

- We provide a structural analysis of isotropy-aware representations, using cosine similarity as a metric to measure isotropy across different contextualized representations. The study reveals that when the cosine similarity is closer to zero, strong isotropy is present in the embedding space which stabilizes the partition function. The stability of the partition function ensures that logits are solely determined by probabilities, thereby resolving the shift-invariance problem of the softmax function and providing a performance guarantee.

- Finally, we present two examples that illustrate the conditions under which isotropy is preserved in LLM representations of numerical data. Our experiments demonstrate that different characteristics of numeric data could have different impacts on the isotropy.

## 1.1 RELATED WORKS

The most relevant works Arora et al. (2016; 2017); Brown et al. (2020); Cai et al. (2021); Gao et al. (2019); Ethayarajh (2019); Rajaee & Pilehvar (2021) are primarily focused the natural language processing (NLP) domain and depend on latent variable model. For instance, in Arora et al. (2016), the latent variable model is employed to explain and unify diverse word embedding algorithms. This theoretical framework is further extended to justify sentence embedding methods, either by leveraging the latent variable model Arora et al. (2017) or through the lens of compressed sensing Arora et al. (2018). Language modeling can also be used to exploit hidden representations, as the probability of the next word is typically computed as the softmax of the product of the hidden representation and the dictionary matrix. Consequently, any zero-shot application of pre-trained autoregressive language models, such as GPT-3 Brown et al. (2020) and T5 Raffel et al. (2020), can be used as a suitable method of exploiting the hidden representations.

Isotropy, on the other hand, often makes the embedding space more effectively utilized and more robust to perturbations, i.e., no extreme directions that can cause numerical instability Ji et al. (2023). In Cai et al. (2021), isotropy is found withing the clusters in the contextual embedding space (i.e., local assessment), as opposed to the previous study of anisotropy caused be the misleading isolated clusters (i.e., global assessment) in Gao et al. (2019); Ethayarajh (2019). Motivated by the local assessment findings in Cai et al. (2021), a local cluster-based method is proposed in Rajaee & Pilehvar (2021) to address the degeneration problem that makes the embedding space less isotropic. However, the prior art in Ji et al. (2023); Cai et al. (2021) relies on the isotropy assessment in the NLP domain only and no assessment has been done so far in numerical domains. Our work is the first to theoretically analyze the efficacy of pre-trained LLMs on numeric downstream through isotropy.

## 2 DATA GENERATION AND MODEL

### 2.1 DATA GENERATION PROCESS

We consider a general data generation process that captures the characteristics of numeric data across various numerical domains. For instance, the primary characteristics of the numeric data are typically dynamic, noisy, time-varying, and often subject to interference. Let $x_t$ be an observation of some numeric data (e.g., stock price, financial index, or received signals from sensors) at time instance $t$. We can express a general numeric numeric data generation process as follows:

$$x_t = [h_t \circ s_t] + n_t, \tag{1}$$

where $s_t$ is the true underlying signal (e.g., true asset price, financial index, or received sensor signals) that we want to observe, $h_t$ represents the environmental or system dynamics that modify the signal (e.g., market volatility or shock, reverberation), $n_t$ is the random disturbance or noise (e.g., random fluctuations or market noise, environmental noise, measurement errors), and $\circ$ is the combination operation, which could be multiplicative, additive or convolution depending on the specific the numerical domain. Moreover, the components $s_t$, $h_t$, and $n_t$ in equation 1 are often time-varying and may follow stochastic processes.

### 2.2 MODEL

**Time series Forecasting.** Similar to next-word prediction by LLMs, the next-value prediction in the numerical domain can be modeled by time series forecasting techniques Jin et al. (2024); Ansari et al. (2024) which are widely adopted in the machine learning literature. Formally, given a time series $\mathbf{x}_{1:T+L} = [x_1, \ldots, x_T, \ldots, x_{T+L}]$, where the first $T$ time instances give the historical context, the next $L$ time instances constitute the forecast region, and the observation of each time instance $x_t \in \mathbb{R}$ is given by equation 1, we are interested in predicting the joint distribution of next $L$ time instances, $p(\mathbf{x}_{T+1:T+L}|\mathbf{x}_{1:T})$. Since, the pre-trained models operate on tokens from a finite vocabulary, using them for time series data requires mapping the observations to a finite set of tokens. Depending on the different fields of numeric applications, various tokenization techniques, e.g., quantization Ansari et al. (2024), patching Jin et al. (2024), among others, can be applied to tokenize the time series and create a time series vocabulary $\mathcal{V}$ of $N$ time series tokens, i.e., $|\mathcal{V}| = N$. Then, the realization of the next $L$ time instances can be obtained by autoregressively sampling from

the predicted distribution $p(\tilde{x}_{T+l+1}|\tilde{\mathbf{x}}_{1:T+l})$, for $l \in \{1, \ldots, L\}$, where $\tilde{\mathbf{x}}_{1:T+l}$ is the tokenized time series. For ease of reading, we express $T + l$ as $T_l$ for the rest of the paper.

**Student Model.** We consider a general pre-trained model for numeric data and open the black box of the pre-trained model at the last layer. Specifically, we assume that the observation probability of $\tilde{x}_{T+1}$ given $\tilde{\mathbf{x}}_{1:T_l}$ satisfies the log-linear model Arora et al. (2016)

$$p(\tilde{x}_{T+1} = j \mid \tilde{\mathbf{x}}_{1:T_l}) \propto \exp(\langle v_{1:T_l}(\tilde{\mathbf{x}}_{1:T_l}), v_j \rangle), \tag{2}$$

where $v_j \in \mathbb{R}^D$ is a vector that only depends on the time series token $j \in \mathcal{V}$, and $v_{1:T_l}(.)$ is a function that encodes the tokenized time series sequence $\tilde{\mathbf{x}}_{1:T_l}$ into a vector in $\mathbb{R}^D$. The log-linear modeling aligns with the commonly used LLMs networks whose last layer is typically a softmax layer.

Let $z_j(\tilde{x}, T_l+1) := \langle v_{1:T_l}(\tilde{\mathbf{x}}_{1:T_l}), v_j \rangle$ be the $j$-th logit and $Z(\tilde{x}, T_l+1) = \sum_{j=1}^N \exp(z_j(\tilde{x}, T_l+1)) = \sum_{j=1}^{|\mathcal{V}|} \exp(\langle v_{1:T_l}(\tilde{\mathbf{x}}_{1:T_l}), v_j \rangle)$ be the partition function Arora et al. (2016), i.e., normalization factor. In LLMs, the partition function is often used to normalize the output probabilities of the model, ensuring that they sum to 1. For our case, we use the partition function $Z(\tilde{x}, T_l + 1)$ to normalize the student model in equation 2 and use this normalized model in training. Then, the normalized student model is given by

$$\forall j \in \mathcal{V}, \quad p(\tilde{x}_{T+1} = j \mid \tilde{\mathbf{x}}_{1:T_l}) = \frac{\exp(z_j(\tilde{x}, T_l + 1))}{Z(\tilde{x}, T_l + 1)} = \frac{\exp(\langle v_{1:T_l}(\tilde{\mathbf{x}}_{1:T_l}), v_j \rangle)}{\sum_{j=1}^{|\mathcal{V}|} \exp(\langle v_{1:T_l}(\tilde{\mathbf{x}}_{1:T_l}), v_j \rangle)}. \tag{3}$$

**Loss Function.** As typical in language models, we use the categorical distribution over the elements in the time series vocabulary $\mathcal{V}$ as the output distribution $p(\tilde{x}_{T+1}|\tilde{\mathbf{x}}_{1:T_l})$, for $l \in \{1, \ldots, L\}$, where $\tilde{\mathbf{x}}_{1:T_l}$ is the tokenized time series. The student model is trained to minimize the cross entropy between the distribution of the tokenized ground truth label and the predicted distribution. The loss function for a single sequence of tokenized time series is given by Ansari et al. (2024); Wu et al. (2023)

$$\ell(v_{1:T_l}) = -\sum_{l=1}^{L+1} \sum_{j=1}^{|\mathcal{V}|} \mathbf{1}_{(\tilde{x}_{T+1}=j)} \log p(\tilde{x}_{T+1} = j \mid \tilde{\mathbf{x}}_{1:T_l})$$

$$= -\sum_{l=1}^{L+1} \sum_{j=1}^{|\mathcal{V}|} \mathcal{D}_{\text{KL}}(p^*(\tilde{x}_{T+1} = j \mid \tilde{\mathbf{x}}_{1:T_l}) \| p(\tilde{x}_{T+1} = j \mid \tilde{\mathbf{x}}_{1:T_l}))$$

$$+ H(p(\tilde{x}_{T+1} = j \mid \tilde{\mathbf{x}}_{1:T_l})), \tag{4}$$

where $p(\tilde{x}_{T+1} = j \mid \tilde{\mathbf{x}}_{1:T_l})$ is the categorical distribution predicted by our student model parametrized by $v_{1:T_l}$, $p^*(\tilde{x}_{T+1} = j \mid \tilde{\mathbf{x}}_{1:T_l})$ is the distribution of ground-truth model, $\mathcal{D}_{\text{KL}}$ is the KL divergence, i.e., weighted log probability difference between the ground-truth and the student model, and $H(p(\tilde{x}_{T+1} = j \mid \tilde{\mathbf{x}}_{1:T_l}))$ is the entropy of distribution $p(\tilde{x}_{T+1} = j \mid \tilde{\mathbf{x}}_{1:T_l})$ which is a constant. Note that our model performs regression via classification Torgo & Gama (1997) through the categorical entropy loss in equation 4. We assume that our student model achieves a small loss value so that the KL-divergence term in equation 4 is also small.

**Numeric Downstream Task.** The numeric downstream task that we are considering is *regression via classification* (as described in the previous section). To define the downstream tasks in the numerical domains, we define the logits of the student model as $\mathbf{z}(\tilde{x}, T_l+1) := \{z_j(\tilde{x}, T_l+1)\}_{j=1}^{|\mathcal{V}|} := \{\langle v_{1:T_l}(\tilde{\mathbf{x}}_{1:T_l}), v_j \rangle\}_{j=1}^{|\mathcal{V}|}$. These logits are just the outputs before the softmax computation and we assume that the numeric downstream task is determined by a function of the logits. Intuitively, $\mathbf{z}$ is the representation learned during pre-training step. A simple numeric downstream task is one whose regression (via classification) is linear in $v_{1:T_l}$, that is, $f(\tilde{x}, T_l + 1) = \langle v_{1:T_l}(\tilde{\mathbf{x}}_{1:T_l}), u^* \rangle = \sum_{j=1}^{|\mathcal{V}|} a_j z_j(\tilde{x}, T_l + 1)$, where $u^* = \sum_{j=1}^{|\mathcal{V}|} a_j v_j \in \mathbb{R}^D$ and $a_j$ denotes the coefficient. This model is still not sufficient to provide a performance guarantee to generalize to numeric downstream task in unseen scenarios. This is due to the fact that, for the entries with small ground-truth probabilities, a large log probability difference does not results in a large KL divergence in the loss function in equation 4. However, the log probability difference is proportional to the difference in the value of the perfect model (i.e., ground-truth) $f^*(\tilde{x}, T_l + 1)$. This allows the student model to alter the

signs of $f^*(\tilde{x}, T_l + 1)$ without resulting in a large KL divergence Wu et al. (2023). Then, it is more reasonable to model the numeric downstream task as

$$f(\tilde{x}, T_l + 1) = \sum_{j=1}^{|\mathcal{V}|} a_j \sigma(z_j(\tilde{x}, T_l + 1) - b_j) = \sum_{j=1}^{|\mathcal{V}|} a_j \sigma(\langle v_{1:T_l}(\tilde{\mathbf{x}}_{1:T_l}), v_j \rangle - b_j), \quad (5)$$

where $\sigma$ is the ReLU function and $b_j$ denotes the threshold for the logits. The numeric downstream task only considers the logits that are above the threshold, and thus ignores all the entries with very small probabilities. However, as we will show in Section 3, the student model in equation 5 still needs to exhibit a structure in LLM hidden representations to provide a performance guarantee for the numeric downstream tasks.

## 3 THE ROLE OF ISOTROPY IN ADAPTING LLMS TO NUMERICAL DATA

### 3.1 NEED OF STRUCTURE IN LEARNED REPRESENTATIONS

**Observation 1:** The hidden representations of LLM embeddings must exhibit structure to address the shift-invariance problem of the softmax function. Without such structure, the model can shift the logits while keeping the training loss unchanged and leaving the logits ineffective for the numeric downstream tasks.

As previously discussed in Section 2.2, we consider LLM networks whose last layer is usually a softmax layer and the numeric downstream task is determined by the function of the logits. The underlying relation between the logits and softmax function determines the performance of the numeric downstream tasks. However, the softmax function is shift-invariant, that is, the output of the softmax function remains unchanged when all logits are shifted by a constant. Since we do not have any control over the logit shift of the student model on unseen data, good performance during training does not necessarily provide any performance guarantee for the numeric downstream task on unseen scenarios. This can be formalized in the following theorem.

**Theorem 1.** *Let assume the logits $\mathbf{z}^*(\tilde{x}, T_l + 1)$ of the ground-truth model is bounded. For any function $f^*(\tilde{x}, T_l + 1) = \sum_{j=1}^{|\mathcal{V}|} a_j \sigma(z_j^*(\tilde{x}, T_l + 1) - b_j)$, there exist functions $\{\hat{z}_j(\tilde{x}, T_l + 1)\}_{j=1}^{|\mathcal{V}|}$ such that for all $\tilde{x}$ and $T_l + 1$, we have $\hat{p}(\tilde{x}_{T_l+1}|\tilde{\mathbf{x}}_{1:T_l}) = p^*(\tilde{x}_{T_l+1}|\tilde{\mathbf{x}}_{1:T_l})$ and $\hat{f}(\tilde{x}, T_l + 1) := \sum_{j=1}^{|\mathcal{V}|} a_j^* \sigma(\hat{z}_j(\tilde{x}, T_l + 1) - b_j^*)$ is always equal to 0. In other words, the pre-training loss of the model $\{\hat{z}_j(\tilde{x}, T_l + 1)\}_{j=1}^{|\mathcal{V}|}$ is the same as the ground-truth model $\{z_j^*(\tilde{x}, T_l + 1)\}_{j=1}^{|\mathcal{V}|}$, but its logits are ineffective for the numeric downstream tasks.*

*Proof.* We choose $\tau \in \mathbb{R}$ such that $\forall \tilde{x}, T_l + 1, \tau < \min_{j \in \mathcal{V}} b_j^* - \max_{j \in \mathcal{V}} z_j^*(\tilde{x}, T_l + 1)$, and $\forall \tilde{x}, T_l + 1, \forall j \in \mathcal{V}$, we set $\hat{z}_j(\tilde{x}, T_l + 1) := z_j^*(\tilde{x}, T_l + 1) + \tau$, then

$$\forall j \in \mathcal{V}, \hat{z}_j(\tilde{x}, T_l + 1) - b_j^* < z_j^*(\tilde{x}, T_l + 1) + \min_{j \in \mathcal{V}} b_j^* - \max_{j \in \mathcal{V}} z_j^*(\tilde{x}, T_l + 1) - b_j^* \leq 0,$$

which implies that $\sigma(\hat{z}_j(\tilde{x}, T_l + 1) - b_j^*) = 0$. Therefore, $\forall \tilde{x}, T_l + 1$, we have $\hat{f}(\tilde{x}, T_l + 1) = 0$. $\square$

Theorem 1 demonstrates that without any structure in the hidden representations of LLM embeddings, the student model is able to shift the logits for any sample while keeping the pre-training loss unchanged. In the worst case scenario, this could lead to drastic shifts in the logits for the unseen data which leads to poor numeric downstream task performance. Consequently, a theoretical guarantee for the numeric downstream task performance needs structure in the LLM representations learned by the pre-trained model.

### 3.2 RELATION BETWEEN ISOTROPY AND STRUCTURE IN REPRESENTATION

**Observation 2:** The isotropic property of LLM embeddings preserves the underlying structure of the representations by approximating the partition function (in equation 3) with a constant for different samples, thereby resolving the shift-invariance problem of softmax function.

One way to prevent the shift-invariance problem from influencing the performance of the numeric downstream tasks is to keep the partition function stable. Note that in equation 3, the probability of a value in any time instance is the exponential of the corresponding logit $z_j(\tilde{x}, T_l + 1)$ divided by the partition function $Z(\tilde{x}, T_l + 1)$. If the partition function remains constant for different samples, the logits can be solely determined by the probabilities, thereby resolving the shift-invariance problem of the softmax function. In the light of the theoretical findings in Arora et al. (2016), if the LLMs' hidden representations are isotropic in contextual embedding space, $Z(\tilde{x}, T_l + 1)$ could be approximated by a constant. Formally, the isotropy of embedding space can be assessed through the partition function $Z(\tilde{x}, T_l + 1)$ Arora et al. (2016); Mu & Viswanath (2018) as follows

$$I(v_{1:T_l}) = \frac{\min Z(\tilde{x}, T_l + 1)}{\max Z(\tilde{x}, T_l + 1)}, \tag{6}$$

where $l = 1, \ldots, L$. Accordingly, when the partition function is constant for different samples, $I(v_{1:T_l})$ would be close to one, indicating for a perfectly isotropic embedding space Arora et al. (2016); Mu & Viswanath (2018). In other words, the isotropic property of LLMs' embeddings in the contextual embedding space preserves the underlying structure of the representation and, thus, makes the logits useful for numeric downstream tasks.

### 3.2.1 STUDY OF ISOTROPY IN LLMS' HIDDEN REPRESENTATIONS

Inspired by Cai et al. (2021), we follow their procedure and study the isotropy in the LLM representations in the contextual embedding space for numeric downstream tasks.

**Clustering.** We begin with the isotropy assessment by performing clustering on the LLMs' representations in the contextual embedding space. There are various methods for performing cultering, such as $k$-means, DBSCAN Ester et al. (1996). We select $K$-means clustering method because it is reasonably fast in high embedding dimensions (e.g., $d \geq 768$ for GPT2, ELMo, BERT etc.). We use the celebrated silhouette score analysis Rousseeuw (1987) to determine the number of clusters $|C|$ in the contextual embedding space. After performing $K$-means clustering, each observation $p$ (i.e., one of the $j$ vector representations in $\mathcal{V}$) is assigned to one of $C$ clusters. For an observation $p$ assigned to the cluster $c \in C$, we compute the silhouette score as follows

$$a(p) = \frac{1}{|C| - 1} \sum_{q \in C, p \neq q} \text{dist}(p, q); \quad b(p) = \min_{\tilde{c} \neq c} \sum_{q \in \tilde{c}} \text{dist}(p, q); \quad s(p) = \frac{b(p) - a(p)}{\max\{b(p), a(p)\}},$$

where $a(p)$ is the mean distance between an observation $p$ and the rest in the same cluster class $p$, while $b(p)$ measures the smallest mean distance from $p$-th observation to all observations in the other cluster class. After computing the silhouette scores $s(p)$ of all observations, a global score is computed by averaging the individual silhouette values, and the partition (with a specific number of clusters) of the largest average score is pronounced superior to other partitions with a different number of clusters. We select the best $|C|$ that belongs to the partition that scores highest among the other partitions.

**Isotropy measurement using cosine similarity metric.** The metric we use for measuring isotropy in LLM representations in the contextual embedding space is cosine similarity metric. Let $k$ be a time series token in time series vocabulary $|\mathcal{V}|$. We call the time series token a *type* for ease of reading. Each type $i$ in $|\mathcal{V}|$ is represented by $k_i$. Let $\Psi(k_i) = \{\psi_1(k_i), \psi_1(k_i), \ldots\}$ be the set of all LLMs' contextual embedding instances of $k_i$ in $\Psi(k_i)$. Note that, different contexts in the different time series sequences yield different LLMs' embeddings of $k_i$. By constructing $\sum_t |\Psi(k_i)| = |\mathcal{V}|$, we define the inter- type cosine similarity as,

$$\zeta_{\cos} \triangleq \mathbb{E}_{i \neq j}[\cos(\psi(k_i), \psi(k_j))], \tag{7}$$

where $\psi(k_i)$ is any random sample from $\Psi(k_i)$, and the same for $\psi(k_j) \in \Psi(k_j)$. The expectation is taken over all pairs of different types. Since each LLMs' contextual embedding instance $\psi(k_i)$ belongs to a particular cluster through clustering, the cosine similarity should be measured after shifting the mean to the origin Mu & Viswanath (2018). Accordingly, we subtract the mean for each cluster (i.e., centroid) and calculate the adjusted $\zeta_{\text{inter}}$. Assuming we have a total of $|C|$ clusters, let $\psi_c(k_i) = \{\psi_c^1(k_i), \psi_c^2(k_i), \ldots\}$ be the set of type $k$'s contextual embeddings in cluster $c \in C$, and $\psi_c(k_i)$ be one random sample in $\psi_c(k_i)$. We define the adjusted inter-type cosine similarity as

$$\zeta'_{\cos} \triangleq \mathbb{E}_c \left[ \mathbb{E}_{i \neq j} \left[ \cos \left( \bar{\psi}_c(k_i), \bar{\psi}_c(k_j) \right) \right] \right], \tag{8}$$

where $\bar{\psi}_c(k_i) = \psi_c(k_i) - \mathbb{E}_{\psi_c}[\psi_c(k_i)]$. Here $\mathbb{E}_c$ is the average over different clusters, and $\bar{\psi}_c(k_i)$ is the original contextual embedding shifted by the mean, with the mean taken over the samples in cluster $c$. The inter-type cosine similarity takes values between $-1$ and $1$. An inter-type cosine similarity value close to 0 indicates strong isotropy and ensures the existence of structure in the LLMs' representations.

# 4 WHEN IS ISOTROPY PRESERVED IN LLM REPRESENTATIONS OF NUMERICAL DATA?

**Analysis settings.** In this section, we present two examples that illustrate the conditions under which isotropy is preserved in LLM representations. To reflect the numeric data generation process in equation 1, we select two datasets of wireless channels as they are dynamic, noisy, time-varying, and subject to interference, and thus hold all primary characteristics of the numerical data across various domains. Specifically, we use time division duplexing (TDD) and frequency division duplexing (FDD) from two different wireless communication settings, as shown in Figure 2. We call the TDD dataset "Dataset 1" and the FDD dataset "Dataset 2". The downstream task here is to predict the channel property using LLM, where Dataset 1 causes good downstream performance, while the Dataset 2 causes bad downstream task performance. We use NMSE as a performance metric for the numeric downstream task because it is widely used for signal prediction. We use the GPT2 Radford et al. (2019) as our pre-trained contextual embedding model and the first six layers of which are deployed. We perform our isotropy evaluations on the pretrained uncased base models from Huggingface (https://huggingface.co/transformers/index.html).

We use the datasets and simulation setups from Liu et al. (2024), which are the standard settings for wireless time series forecasting. We predict $L = 4$ future channel properties based on the historical $T = 16$ Channel properties through time series forecasting using GPT2. The training and validation dataset contains $8,000$ and $1,000$ samples, respectively, with user velocities uniformly distributed between 10 km/hour and 100 km/hour. The test dataset contains ten velocities ranging from 10 km/hour to 100 km/hour, with $1,024$ samples for each velocity.

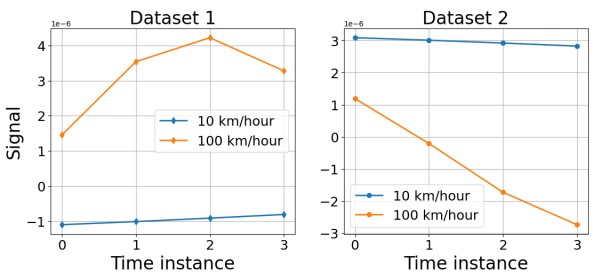

Figure 2: Visualization of dataset 1 (Left) and dataset 2 (Right) labels for two extreme cases of velocities, i.e., 10 km/hour and 100 km/hour.

## 4.1 EXAMPLE 1: ISOTROPY IN DATASET 1

**Performance of Dataset 1.** In this section, we provide an example of a good numeric downstream task performance with Dataset 1. For instance, in Figure 3, we compare the NMSE performance of our GPT2 based channel prediction model with baselines for different user velocities. From Figure 3, we can observe that the NMSE performance of all baselines gradually increased along with the increase in user velocity. This is because, with the increase in velocity, the wireless channel characteristics rapidly changes within a very short coherence time, resulting in increased prediction difficulty for the prediction model. The GPT2 based model consistently outperforms other baselines and demonstrates its high prediction accuracy.

**Effective dimension in Dataset 1.** For Dataset 1, we first analyze the effective GPT2 embedding dimensions through PCA. There are $D = 768$ embedding dimensions for GPT2. For each layer of GPT2, we start with the data matrix, $M \in \mathbb{R}^N$, where $N$ is the number of input tokens and $M$ is the original number of dimensions. We perform PCA to reduce the embedding dimension and project the original Dataset 1 into a 3-D view in Figure 4, with 50 km/hour user velocity. Let the explained variance ratio be: $r_m = \sum_{i=0}^{m-1} \sigma_m / \sum_{i=0}^{d-1} \sigma_m$, where $\sigma_i$ is the $i$-th largest eigen value of covariance matric of $M$.

We are particularly interested in the last layer (i.e., layer 6) as it is related to the logits $\mathbf{z}$ we used in our model in equation 3. From Figure 4, we can observe the first three principal components account for 76% of the total variance in layer 6. Also, we can see from Figure 4 that there are two disconnected islands that are far away from each other in layer 6. When the variance (i.e., $r_m$) is dominated by the distances between clusters, the isotropy estimation would be biased by the inter-cluster system. In this case, it is more reasonable to consider a per-cluster study of isotropy rather than a global estimate.

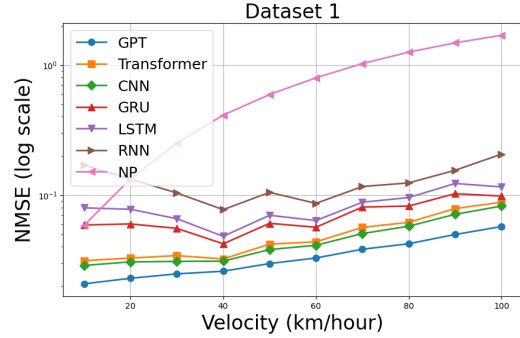

Figure 3: GPT2 outperforms all other baselines for all of the ten different velocities for Dataset 1.

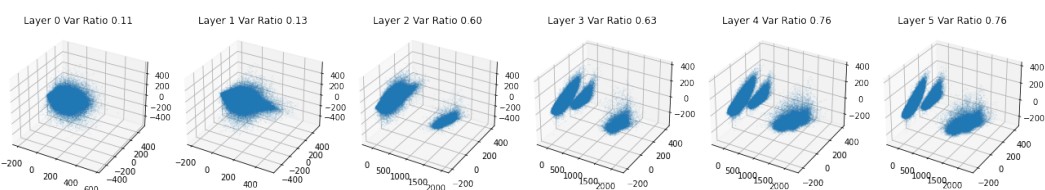

Figure 4: Illustration of effective dimension using PCA for TDD dataset. Isolated clusters exist in the GPT2's contextual embedding space from layer 3 to layer 6.

**Isotropy assessment for Dataset 1.** For illustrative purposes, we pick three user velocities: 10 km/hour, 50 km/hour, and 100 km/hour, for isotropy assessment of Dataset 1. The GPT2 based model achieves good NMSE performance for all of these three velocities, as shown in Figure 3. We perform the clustering by $K$-means clustering, as it is more reasonable for the Dataset 1. We apply adjusted inter-type cosine similarity $\zeta'_{\cos}$ (as in equation 8) to measure the isotropy in GPT2 embedding space. From Figure 5, we can see that the GPT2 based model has consistent near-zero cosine similarity values for all layers, including layer 6. This indicates that nearly perfect isotropy exists in the GPT2 embedding space for the Dataset 1, which preserves the structure in the GPT2's hidden representations and causes good downstream task performance.

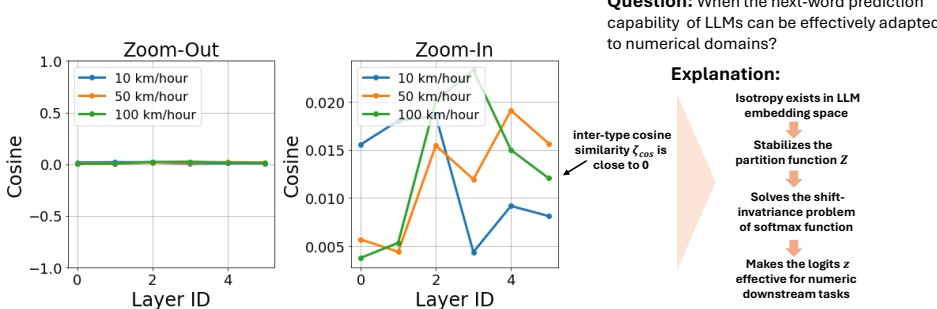

Figure 5: Inter-type cosine similarities for Dataset 1 with different velocities. $\zeta'_{\cos}$ are close to zero for all the layers, including layer 6, indicating that nearly perfect isotropy exists in the GPT2 embedding space for the Dataset 1, which preserves the structure in the GPT2's hidden representations and causes good downstream task performance.

### 4.2 EXAMPLE 2: ISOTROPY IN DATASET 2

**Performance of Dataset 2.** In this section, we provide an example of bad numeric downstream task performance. As shown in Figure 6, the NMSE per-

formance fluctuates randomly for different velocities, while the NMSE performance for Dataset 1 is gradually increasing with increase in the velocities. The NMSE performance for Dataset 2 deteriorates significantly compared to the Dataset 1.

**Effective dimension in Dataset 2.** In Figure 7, analyze the effective GPT2 embedding dimensions for Dataset 2 through PCA. As before, we are particularly interested in the last layer (i.e., layer 6) as it is related to the logits $\mathbf{Z}$. From Figure 7, we can observe the the first three principal components account for $r_m = 92\%$ of the total variance. We can also see that there are no separated islands, as we see for the Dataset 1. Hence, it is more meaningful to consider a global estimate of isotropy for Dataset 2, as opposed to performing per cluster investigation.

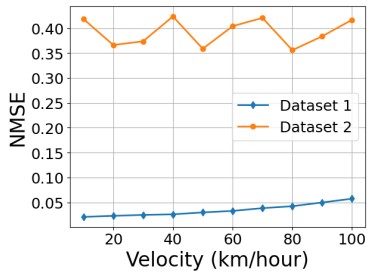

Figure 6: The NMSE performance of the GPT2 based time series forecasting for Dataset 2 deteriorates significantly compared to Dataset 1.

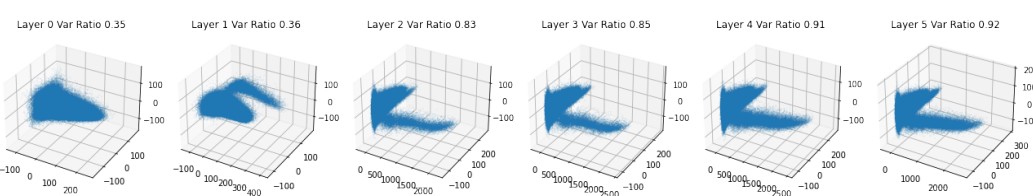

Figure 7: Unlike Dataset 1, no isolated clusters exist in the GPT2's embedding for Dataset 2.

**Isotropy assessment for Dataset 2.** As before, with the three user velocities, the NMSE performance for Dataset 2 for all of these velocities is worse as compared to Dataset 1, as shown in Figure 6. From Figure 8, we can observe a weak isotropy (i.e., anisotropy) in the LLM embedding space for Dataset 2, cauing a lack of structure in the GPT2 hidden representations, and thus leading to bad downstream performance.

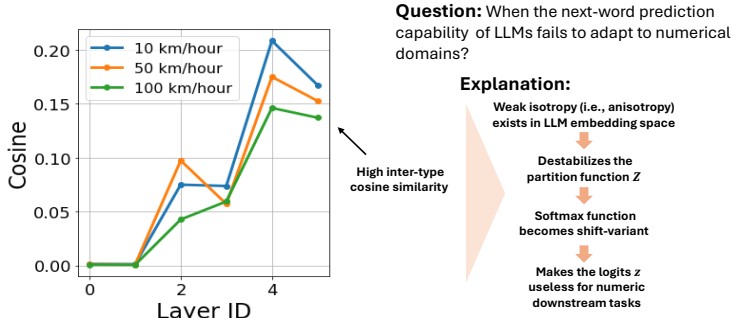

Figure 8: Inter-type cosine similarities for Dataset 2 for different velocities. Higher $\zeta_{\cos}$ values indicate a weak isotropy (i.e., anisotropy) exists in the LLM embedding space which causes a lack of structure in the GPT2 hidden representations, yielding bad downstream performance.

## 5 CONCLUSION AND LIMITATIONS

Isotropy in embeddings as studied here can serve as a foundation for future research on the deeper understanding of LLMs and their applications in various domains. Beyond isotropy, there could be other methods to approximate the partition function with a constant and make the logits useful for the numeric downstream tasks. Moreover, our isotropy study only ensured the existence of structure in the LLMs' hidden representations and provides a performance guarantee when the structure is preserved by isotropy. Improving the numeric downstream task performance when structure is not preserved in the LLM representations is a topic of future work.

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
