# OpenReview forum: "When can isotropy help adapt LLMs' next word prediction to numerical domains?"
_ICLR.cc/2025/Conference — ICLR 2025 Conference Withdrawn Submission_

### Official Review · Reviewer_F5SX · 2024-11-02

**Soundness:** 2
**Presentation:** 2
**Contribution:** 3
**Rating:** 3
**Confidence:** 4

**Summary:**

The paper addresses the adaptation of LLMs for numerical downstream tasks through the lens of isotropy in embedding representations. It provides a theoretical framework to understand when and how LLMs' next-word prediction capabilities can be effectively harnessed for numerical data. This helps improve the interpretability of this field.

**Strengths:**

This paper introduces a novel explanatory perspective by exploring the role of isotropy in the embedding spaces of LLMs adapted for numerical predictions. And the author attempts to give a theoretical proof for his idea.

**Weaknesses:**

1. The paper suffers from poor readability and overcomplication of concepts. For instance, Contribution 2 introduces a theorem (Theorem 1.) to prove "why the shift-invariance problem needs to be addressed," which I believe is entirely unnecessary.

2. I do not understand the rationale for framing regression problems as classification tasks, even though it is technically feasible. Additionally, the authors did not provide any code, and it is unclear how the time series vocabulary (mentioned in line 161) is constructed. If it uses the GPT-2 vocabulary, which is composed of numbers, how are the basic units of the time series vocabulary determined?

3. It is also unclear why a "Student Model" is suddenly introduced without explaining what the "Teacher Model" is.

4. Regarding the experimental section, I am puzzled as to why the authors chose to simulate data for wireless communications, especially since the paper mentions "applications in finance, energy, retail, climate science, wireless networks, and synthetic tabular generation, among others". Why not analyze using open-source datasets readily available for these applications?

5. The paper always uses $T_l+1$, but I think it should be $T_{l+1}$. Concerning Eq.4, I believe there is a significant issue: KL divergence (relative entropy) is inherently an expectation, so why is there an additional summation outside? Also, is there a problem with the entropy term H?

6. Section 3.2, which I initially thought would be the most valuable, is not clearly explained by the authors. It quickly transitions to measuring isotropy and clustering, which largely reuses the analytical methods proposed by Cai[1].

7. Furthermore, there are formatting issues such as improper line breaks at line 403 and incorrect spacing between lines 431-433. Additionally, all figures seem to lack clear annotations, giving the impression that the paper was hastily prepared.

[1]Cai X, Huang J, Bian Y, et al. Isotropy in the contextual embedding space: Clusters and manifolds[C]//International conference on learning representations. 2021.

**Questions:**

Please refer to weakness.

---

### Official Review · Reviewer_Fnsz · 2024-11-02

**Soundness:** 2
**Presentation:** 2
**Contribution:** 3
**Rating:** 5
**Confidence:** 3

**Summary:**

The authors address the following question: when can next-token capabilities of LLMs be effectively adapted to numerical domains?
The authors respond by claiming that next-token capabilities can be extended to numerical domains when there there is isotropy in the LM contextual embeddings. Essentially, if your numerical data is a function of the logits, the model performance might not reflect an accurate representation of the numerical data since the post-softmax results may lose information due to the shift-invariance of the softmax function. The authors demonstrate experimental results with GPT on two time series datasets.

**Strengths:**

- The study is well motivated in trying to understand when next token prediction capabilities of LLMs will extend to numerical data
- The authors provide a plausible argument for the role of isotropy in adapting LLMs to numerical data which builds on prior work

**Weaknesses:**

The experimental results are not very extensive; the main experiment compares performance of a GPT model on two time series datasets and shows that the model performs better on Dataset 1, in which case the model learns isotropic representations, than Dataset 2, in which case the model does not learn isotropic representations. This seems to suggest there is some underlying property of the data that is determining the performance (the existence of isotropic representations does not necessarily seem causal). It would be interesting to see a setting in which multiple models (with some variation such as architecture) are trained on the same dataset and learn representations with varying levels of isotropy, and testing whether the level of isotropy corresponds to performance.

**Questions:**

-  I am not very familiar with the numerical data setting -- Why is the shift-invariance of softmax specifically a problem for numerical data as compared to other downstream tasks?
- Can experimental results be shown across more extensive settings? Ie, with more empirical evidence, can isotropy be shown to be a causal factor in performance? Or otherwise, is there some property of the data that can predict whether the model will learn an isotropic representation?

---

### Official Review · Reviewer_JV8A · 2024-11-03

**Soundness:** 3
**Presentation:** 2
**Contribution:** 2
**Rating:** 3
**Confidence:** 3

**Summary:**

This paper studies that, after training a LLMs on next-word prediction, if an application needs to adapt the pre-softmax LLM output for a numerical task, whether isotropy helps. The authors identify that the shift-invariance property of the softmax function would cause problems for such applications. They conduct analyses and experiments on two datasets (TDD and FDD) and the results seem to match their intuition.

**Strengths:**

The analyses and the intuition (for this particularly scenario) are reasonable.

**Weaknesses:**

1. The paper is not well written;
2. Based on my knowledge, the setting studied in this paper is not very popular, and its not well argued that this is an important problem or may impact a wide range of applications.
3. The experiments are conducted with relatively weak model (GPT-2) and no experiments are conducted on popular benchmarks, making it hard to judge the significance of the observations discussed in this paper.

**Questions:**

1. As to the two datasets used (TDD & FDD), what are their settings and how are they collected? Citations and discussions are needed for these information. Since these two seems to be wireless network datasets, which is not common knowledge for most ICLR audiance.

2. This paper seems to target the numerical domain application in general (in the introduction, it lists several domains including finance, energy, retail, climate, etc.), is there evidence that the setting described in this paper is widely used in these domains? If so, I'm wondering whether you can add more experiments from these domains as well?

3. The choice of many terminology in this paper is not very popular and is even a bit confusing. For example, why it is "Student Model" in line-165? Note that "student model" is widely used in knowledge distillation.

---

### Official Review · Reviewer_AfTK · 2024-11-04

**Soundness:** 2
**Presentation:** 2
**Contribution:** 1
**Rating:** 3
**Confidence:** 3

**Summary:**

This paper explores how structure in pretrained LLM embeddings is important for using these models for numerical domains tasks (such as time series forecasting). Specifically, the authors propose that when isotropy in the embeddings preserves the structure in LLM representations, it resolves the shift-invariance problem in the model's softmax function (i.e., the softmax output remains unchanged when all logits are shifted by a constant). They validate this hypothesis through examples with synthetic datasets, demonstrating cases where isotropy improves or hinders downstream performance.

**Strengths:**

- The paper provides a new perspective on LLM embedding structure in the context of time series analyses. The method of analyzing performance relative to isotropy is a new and potentially interesting avenue of exploration.
- The interdisciplinary nature of the paper (exploring LLM applications in settings outside of the scope of NLP) is timely.
- The paper is well written.

**Weaknesses:**

- The paper is very limited in its scope of analysis. It presents results on only two synthetic datasets that limits the generalizability of findings. Moreover, the apparent difference in performance on the two datasets is purported to be because of a difference in isotropy in embeddings, but the causal link is not actually shown. Attributing poor performance on Dataset 2 to low isotropy seems speculative without exploring other possible causes.
- The theoretical claims of the paper are largely qualitative, lacking rigorous, quantitative backing, and due to the limited scope of analysis, it is hard to know exactly when and where these results hold.
-Too much space is dedicated to background ideas (such as the noisy data generation process of time series data), thereby limiting room for original contributions.

**Questions:**

- Can the authors provide more details on the datasets that are used in this paper, as well as their application to real-world use cases?

---

### Note · Authors · 2025-01-30

I have read and agree with the venue's withdrawal policy on behalf of myself and my co-authors.